# Acute Effects of Kickboxing K1 Matches on Hematological Parameters of Kickboxers

**DOI:** 10.3390/jfmk9030130

**Published:** 2024-07-26

**Authors:** Marta Niewczas, İsmail İlbak, Serkan Düz, Tomasz Pałka, Tadeusz Ambroży, Henryk Duda, Wojciech Wąsacz, Paweł Król, Robert Czaja, Łukasz Rydzik

**Affiliations:** 1Institute of Physical Culture Studies, College of Medical Sciences, University of Rzeszow, 35-959 Rzeszow, Poland; martaniewczas@wp.pl (M.N.); pkrol@ur.edu.pl (P.K.); rczaja@ur.edu.pl (R.C.); 2Institute of Health Sciences, İnönü University, 44000 Malatya, Türkiye; isma_ilbak@hotmail.com; 3Faculty of Sports Sciences, Coaching Education, İnönü University, 44000 Malatya, Türkiye; serkan.duz@inonu.edu.tr; 4Department of Physiology and Biochemistry, Faculty of Physical Education and Sport, University of Physical Education in Krakow, 31-571 Krakow, Poland; tomasz.palka@awf.krakow.pl; 5Institute of Sports Sciences, University of Physical Education, 31-571 Krakow, Poland; tadek@ambrozy.pl (T.A.); henryk.duda@awf.krakow.pl (H.D.); wojciech.wasacz@doctoral.awf.krakow.pl (W.W.)

**Keywords:** kickboxing K1, hematological parameters, acute effects, athletes, physiological changes

## Abstract

While there is clear evidence in the literature that the hematological parameters in athletes of different sports are affected by exercise and varying loads, to our knowledge, there are limited studies on the real impact of kickboxing matches on kickboxers’ hematological parameters. In this context, this cross-sectional study was conducted to examine the acute changes in the hematological parameters of kickboxers following K1 matches. With the participation of 10 kickboxing K1 athletes, the hematological parameters, including the WBC, Plt, Neut, Lymph, Mono, RBC, Hgb, Hct, CK, La, and glucose levels, were examined before and after matches. Paired sample *t*-tests were used to compare the pre-test and post-test hematological parameters of the participants. The findings indicated statistically significant differences in the post-match WBC, Plt, Neut, Lymph, CK, La, and glucose levels, while no statistically significant differences were observed in the RBC, Hct, Hgb, and CK levels (*p* < 0.05). These results not only emphasize the complexity of physiological changes in athletes, but also show consistency with various findings in the literature, while contradicting some. Therefore, it is highlighted that further research is needed to understand the effects of K1 matches on hematological parameters.

## 1. Introduction

Kickboxing is a combat sport that requires physical endurance, speed, muscle power, and technical skills [1]. Characterized by constantly changing movement patterns involving submaximal and maximal loads [2], it is characterized by varied intensities and durations of exertion [3,4]. Particularly, K1 matches entail fast-paced, physically intense encounters often consisting of frequent and short rounds. Given that such competitions push athletes to their physical and physiological limits [5], understanding how hematological parameters are affected by this intense activity is of paramount importance for athletes’ health and performance.

Despite the available literature, the knowledge regarding the acute effects of kickboxing K1 matches on kickboxers’ hematological parameters is quite limited. The existing studies in this field have generally focused on different sports or varied intensities of training. In this context, Azarbayjani [6] reported a significant increase in white blood cell (WBC), lymphocytes (Lymph), and platelet (Plt) counts in kickboxers following a single session of aerobic exercise comprising three sets of three-minute rest periods. Ouergui et al. [7] reported elevated glucose and lactate concentrations (La) in athletes after a simulated kickboxing match. Hazratian et al. [8] found that single-session intensive judo exercises led to significant post-exercise decreases in the red blood cell (RBC), hemoglobin (Hgb), hematocrit (Hct), and mean corpuscular hemoglobin concentration (MCHC) values in elite adolescent judokas, with the mean corpuscular hemoglobin (MCH) and red distribution width (RDW) values remaining unchanged. Hebisz et al. [9] identified increases in the RBC, WBC, Hgb, and Hct values in endurance athletes following repeated sprint exercises compared to the baseline. Furthermore, Boyalı et al. [10] reported decreases in the Hct, mean corpuscular volume (MCV), MCH, mean platelet volume (MPV), and plateletcrit (Pct) values in elite taekwondo athletes after an eight-week preparatory exercise period.

While there is clear evidence in the literature that the hematological parameters of athletes from different sports are affected by exercise and varying intensity loads [6,11,12,13,14], to our knowledge, there are limited studies on the real impact of kickboxing matches on kickboxers’ hematological parameters.

Understanding the effects of an official kickboxing K1 match on certain blood parameters of athletes can facilitate the development of more effective approaches in training programs and pre- and post-competition care strategies. Additionally, this paper can provide significant clinical guidance for kickboxing coaches, sports physicians, and other relevant health professionals, thereby enhancing athletes’ health and performance. This study also offers a comprehensive framework for understanding the extent of physiological stress experienced by athletes, which could benefit other athletes subjected to similar intense physical activity.

Therefore, the aim of this study was to examine the acute changes in hematological parameters (WBC, Plt, Neut, Lymph, Mono, RBC, Hgb, Hct, CK, La, and glucose) in kickboxers following an official kickboxing K1 match. The primary hypothesis of this research was that an official kickboxing K1 match would induce acute changes in the blood parameters of the athletes.

## 2. Materials and Methods

### 2.1. Participants

The minimum sample size for this study was calculated using G*Power software (version: 3.1.9.7; Dusseldorf University, Dusseldorf, Germany). Accordingly, *t*-tests were used to calculate the power according to the study design. Means: difference between two dependent means; α error probability = 0.05; effect size = 0.90; and power (1-β error probability) = 0.80 were determined. Based on this analysis, the minimum sample size required for statistical significance determined by the software with a real power of 0.83% was understood to be at least 10 participants. Therefore, the group of participants in this study included 10 kickboxers. The group was determined using the homogeneous sampling technique within the scope of the purposeful sampling method. In this context, the inclusion criteria of the study were determined as follows: being a competitive kickboxing K1 athlete for a minimum of 5 years, having no health problems, not using any anabolic agents, and voluntary participation in this research. The exclusion criteria were being a kickboxing athlete for less than 5 years, having any injuries or health issues, and using anabolic agents or illicit drugs.

All the participants underwent anthropometric measurements following the techniques and standards recommended by the International Society for the Advancement of Kinanthropometry (ISAK) [15]. In this regard, body height measurements were taken barefoot using a stadiometer (SECA, Hamburg, Germany) with a precision of 0.01 m, whereas body weight (BW) was measured only in shorts using an electronic scale (Tanita, SC-330, Tokyo, Japan) with a precision of 0.1 kg. The anthropometric measurements and demographic information of the participants are presented in Table 1.

### 2.2. Experimental Design

This cross-sectional study was conducted in accordance with the Declaration of Helsinki after obtaining approval from the Ethics Committee of the Regional Medical Board in Kraków (Approval No. 226/KBL/OIL/2023). Before this research, all the participants provided informed consent by signing a consent form and declaring their voluntary participation. Demographic information, including age, body height, and body weight, was collected from all the participants. To determine the effects of kickboxing K1 matches on the athletes’ hematological parameters (WBC, Plt, Neut, Lymph, Mono, RBC, Hgb, Hct, CK, La, and glucose), venous blood samples were collected before and after the matches (2 min post-match) using the venipuncture method and transferred to biochemical test tubes. The samples were then analyzed at the laboratory after cold chain transportation.

### 2.3. K1 Match

The kickboxing bouts, organized in accordance with the K1 rules, took place in a 6 × 6 m ring and consisted of three rounds, each lasting 2 min. There was a 1-min break between rounds. During the fight, competitors were wearing 10 Oz gloves, protective helmets, shin guards, and foot protectors, as per the rules of the World Association of Kickboxing Organization (WAKO). The entire match was supervised by a ring referee.

### 2.4. Blood Collection and Biochemical Analyses

Venous blood samples were collected from the participants by an experienced specialist using the venipuncture method 2 min before (resting level) and 2 min after the match (acute effect of the match). A volume of 5 mL of blood was drawn during each blood collection. The collected blood samples were transferred to EDTA tubes for hematological analysis and biochemical test tubes for serum separation, maintaining cold chain transportation to ensure sample integrity during transit to the laboratory.

For hematological analysis, the blood samples in EDTA tubes were analyzed using a Sysmex automatic hematology analyzer (Model K-1000, Sysmex Corporation, Kobe, Japan), which determines morphotic elements in whole blood, including white blood cells (WBC), platelets (Plt), neutrophils (Neut), lymphocytes (Lymph), monocytes (Mono), red blood cells (RBC), hemoglobin (Hgb), and hematocrit (Hct).

The blood samples in the biochemistry tubes were centrifuged at 4000 rpm for 10 min at +4 °C to separate the sera. The serum samples obtained were then stored at −40 °C until biochemical analyses were performed. For analysis, the samples were thawed at room temperature. Serum levels of creatine kinase (CK), lactate (La), and glucose were measured using appropriate biochemical analyzers (Beckman Coulter AU480, ABD, Brea, CA, USA).

### 2.5. Statistical Analysis

The data obtained in the study were analyzed using the IBM Statistics package program (SPSS version 25.0, Armonk, NY, USA). After testing the normality of the data using skewness and kurtosis values (±1.5 SD) [16,17,18], it was determined that the data showed normal distribution. Paired sample *t*-tests were used to compare the pre-test and post-test hematological parameters of the participants. The results were evaluated at a significance level of *p* < 0.05 with a 95% confidence interval.

## 3. Results

The findings obtained in this research are presented in Table 2.

According to Table 2, statistically significant differences were observed in the WBC, Plt, Neut, Lymph, CK, La, and glucose levels following the K1 kickboxing match. In addition, based on expert observation, no differences were found between the winner and the loser.

## 4. Discussion

This study was conducted to examine the acute changes in hematological parameters in kickboxers after an official K1 competition. The research findings showed a statistically significant increase in the athletes’ post-competition WBC, Plt, Neut, Lymph, Mono, CK, La, and glucose levels, but there were no significant differences in the RBC, Hgb, and Hct levels. Since the number of direct studies on this topic is quite limited, the research findings were evaluated in conjunction with other studies that examined the changes in various blood parameters after combat sports bouts or high-intensity exercise.

The study findings revealed a significant increase in the number of total WBC, Neut, Lymph, and Mono after the K1 match. Belviranli et al. [19], Ghanbari-Niaki et al. [20], Lambert et al. [21], and Pal et al. [22] reported similar results, whereas the findings of Pitsavos et al. [23] and Ramos-Campo et al. [24] were different. Acute stress induced by exercise is recognized to elevate the WBC count. Nevertheless, the extent of this increase in WBC count following acute exercise is contingent upon the intensity and duration of physical activity [25,26,27]. İbiş et al. [28] reported no significant changes in the hematological parameters after aerobic exercise, but significant increases in the WBC values immediately after anaerobic exercise. In contrast, Çakmakçi [29] reported no significant change in the WBC values of taekwondo athletes before and after a 4-week national team training camp. Another study found that a single session of high-intensity judo exercises had no effect on the WBC count [8].

The increase in WBC concentration following exercise can be attributed, in part, to the reduction in plasma volume. It is thought that the decrease in plasma volume during short-term exercise may occur due to the shift of fluid into the interstitial spaces, increased sweating, and elevated blood pressure, which may concentrate blood components such as the white blood cells in bulk. In addition, although it is recognized that the WBC count rises after activities inducing muscle fatigue, the precise mechanisms governing its behavior across different types of exercise remain incompletely understood. Additionally, some WBCs may serve as part of the body’s defense system, migrating towards muscle fibers to aid in repairing damage incurred during physical activity. Moreover, the elevation in WBC count could be attributed to their role in combating disorders affecting the natural function of certain body tissues [30]. Mechanical factors, such as heightened blood flow rate due to exercise, may also contribute to the observed increase in WBC count [31,32].

The fluctuations in WBC count during short-term activities lasting less than an hour may be attributed to an elevation in the epinephrine hormone levels. This phenomenon is supported by the understanding that catecholamines and cortisol typically induce a decrease in the WBC counts during prolonged activities [32]. Additionally, Zhang et al. [33] demonstrated a significant increase in WBC and Lymph counts during exercises lasting less than 30 min. These conflicting findings across studies conducted in various sports are believed to stem from differences in activity type, intensity, and duration.

Neves et al. [25] reported that high-intensity acute exercise led to a temporary increase in the number of Lymph and Mono, a finding corroborated by Pal et al. [22] who observed increased counts of Neut, Lymph, and Mono following exercise. These outcomes align with the findings of the present study. The elevated WBC count following acute exercise primarily arises from the increased the circulation of Neut, Lymph, and Mono due to exercise-induced stress [34,35]. The evidence suggests that the surge in NEUT after acute exercise results from the inflammatory response associated with exercise-induced skeletal muscle damage and cardiac stress [36]. Another proposed explanation for the post-exercise increase in Lymph count is the heightened release of epinephrine and norepinephrine [37]. Anaerobic and resistance exercises are found to stimulate greater Lymph production compared to that of other sporting activities, attributed to the activation of sympathetic and beta-adrenergic pathways. Sympathetic nerves can directly influence the spleen, thymus, and lymph nodes. Although it has not been directly measured, it is recognized that increased catecholamine and cortisol concentrations contribute to Lymph production [38,39].

The findings of this study revealed a significant increase in the Plt count following the K1 match. This finding aligns with the results reported by Heidari et al. [40] and Okeke et al. [41]. It is known that the level of Plt varies according to many factors, such as the intensity and duration of activity, as well as the participants’ fitness level [42]. Similar to the study findings reported by Hazratian et al. [8], which indicated no change in Plt concentration in response to high-intensity acute judo exercise in adolescent elite judokas, there are other studies with comparable results [43,44]. In contrast to these studies, an increase in Plt concentrations has been observed in elite athletes in response to half-marathon (21.1 km) and high-intensity aerobic interval training (approximately 80% of VO_2_max) [45,46]. Wardyn et al. [47] reported a significant increase in the concentration of Plt percentages. Significant increases in the number of Plt were reported as a result of exhaustive aerobic exercise in a group of young male and female sedentary individuals [34,35,42,48]. Since the platelet cycle is heterogeneous in terms of size, density, and reflectivity, it can be concluded that the age and size of platelets are independent determinants.

The observed post-exercise increase in Plt count can be attributed to hemoconcentration and elevated levels of epinephrine, which stimulates Plt release from storage sites, such as the spleen, the liver, and the lungs [49,50]. Epinephrine release induces strong contractions in the spleen, where approximately one-third of the body’s stored Plt resides, potentially explaining the increase in PLT during exercise. Reviewing the literature reveals variability in hematological parameters immediately following acute exercise, influenced by factors such as exercise type, duration, intensity, and athlete’s nutritional status [34,35,42,48]. There is no consensus regarding the effects of exercise on Plt concentration; however, short-term exercise is known to increase Plt production [31,38]. To elucidate the factors contributing to the increase in Plt count after K1 matches, a detailed examination of hematological and hormonal parameters during the recovery period is necessary.

The results of the study showed no statistically significant difference in the RBC, Hct, and Hgb levels after the K1 competition. Similarly, Azarbayjani et al. [6] reported no significant changes in the number of Hgb and Htc in their study of kickboxers. While these findings are similar to those of Mairbäurl [51] and Pal et al. [22], they differ from the findings of Brun [52]. Wardyn et al. [47] reported a significant increase in the concentration of Htc, and Hgb. Hazratian et al. [8] reported that high-intensity acute judo exercise led to a decrease in Hct levels in adolescent elite judokas. Additionally, it has been reported that 8 weeks of taekwondo training during the preparatory period also resulted in a decrease in Hct levels [10]. In this study, no statistically significant differences in Hct values were found after kickboxing matches according to the research findings. Boyalı et al. [10] suggested that these fluctuations in Hct values may be attributed to plasma losses induced by exercise or that the decrease in erythrocyte values may be due to intravenous hemolysis resulting from intense exercise-induced trauma.

Although no changes in Hgb levels were found in this research, there are studies reporting that Hgb levels are higher after high-intensity exercise compared to those before exercise [53,54]. Increased metabolic processes due to high-intensity exercise lead to the increased demand for oxygen. It is expected that there would be an increase in hemoglobin levels, which plays a crucial role in transporting and delivering oxygen to cells, during exercise compared to those at rest [55]. The inconsistency between the research findings and the findings in the literature may be associated with the insufficient 2-min rest period given after the match. Moreover, there is no consensus yet on whether Hgb levels increase after acute exercise. Hazratian et al. [8] reported a decrease in Hgb levels after acute exercise in judokas in a study. Consistent with the results of this study, it has been shown that a session of resistance exercises significantly decreases the Hgb levels immediately afterward [56]. However, another study found no significant difference in Hgb levels measured before and after acute resistance training between athletes and sedentary males [42]. Therefore, more research is needed to fully understand the effects of acute exercise on Hgb.

Upon reviewing the literature, studies were identified demonstrating significant decreases in the number of RBC following anaerobic exercise [6,8,54]. The immediate decrease in RBCs post-exercise can be attributed to short-term hemoconcentration induced by plasma loss during a match. The potential change in plasma volume without altering hematocrit can be explained by the immediate release of stored red blood cells from the spleen, which maintains the ratio of red blood cells to plasma. Moreover, it is recognized that hypoxic conditions serve as a stimulant for erythropoietin (RBC) production. Furthermore, the decrease in RBC count is attributed to the compression of erythrocytes passing through vessels surrounding the contracting skeletal muscles as a result of high-intensity physical exertion. In other words, this compression is thought to potentially induce erythrocyte rupture, leading to a decrease in the number of erythrocytes in athletes’ circulatory systems [52]. The discrepancy between the research findings and these observations may stem from athletes intermittently compressing blood vessels during maximal muscle contraction only during strikes, aiming to both strike more forcefully and use their energy more efficiently to gain an advantage over their opponents.

The significant increase observed in glucose and La levels following an official kickboxing K1 match in the study is supported by existing research in the literature. Ouergui et al. [7] reported an increase in athletes’ glucose and La levels following a simulated kickboxing match. The continuous increase in glucose during kickboxing matches has been associated with the initiation of gluconeogenesis in the liver due to increased cortisol activation [57]. However, Battezzati et al. [58] mentioned that cortisol is not the sole hormone responsible for increased glucose release and indicated that epinephrine is the most important counter-regulatory hormone capable of enhancing glycogenolysis and gluconeogenesis. On the other hand, the increase in La levels is linked to the significant reliance on anaerobic glycolysis during matches [7], as well as with the predominant energy system utilized during exercise, as reported in the other studies [59]. This study concluded that the post-match increase in glucose and La levels was likely due to the reasons mentioned above.

As the CK molecule is one of the most common serum markers of muscle injury [60,61], the statistical changes observed in CK levels in our study indicate that a K1 kickboxing competition induces muscle damage in athletes. There are similar studies in the literature supporting this finding. In this context, Poderoso de Souza et al. [62] reported an increase in CK levels in athletes following an MMA fight. Additionally, these findings are consistent with the observations by Brandão et al. [63], who noted an increase in CK levels in athletes after a jiu-jitsu competition. The rise in CK levels reflects muscle damage in active muscles, indicating that the integrity and composition of the plasma membrane are affected [64].

Coaches and athletes can use these findings to better understand the immediate post-competition physiological responses in kickboxers. This knowledge can aid in optimizing recovery strategies and adjusting training programs to mitigate the potential negative impacts on health and performance. Knowing that certain hematological parameters like the WBC, PLT, and glucose levels increase significantly post-match, coaches can tailor training sessions and recovery protocols accordingly. For instance, emphasizing recovery strategies that address these acute changes, such as nutrition and hydration strategies, can potentially enhance performance and reduce injury risks.

This study has several limitations. Firstly, the participant group consisted solely of male individuals. Secondly, the sample size was limited to 10 participants. Thirdly, only the following blood parameters were examined: WBC, Plt, Neut, Lymph, Mono, RBC, Hgb, Hct, CK, La, and glucose. Fourthly, the total energy expenditure of athletes during the competition was not calculated. Fifthly, the hematological parameters were not evaluated during the recovery period. Additionally, this study did not include the analysis of any hormonal parameters, which is another limitation of this research in the field of sports sciences.

## 5. Conclusions

This study found significant increases in the WBC, Plt, Neut, and Lymph levels following a K1 match, indicating substantial physiological responses among kickboxers. Furthermore, elevated CK levels were recorded, suggesting potential muscle damage and the early stages of repair. However, it is important to note that assessing CK levels shortly after exertion (2 min) may not accurately reflect the peak levels. The increases in La levels reflect heightened anaerobic metabolism, while elevated glucose levels likely indicate increased energy demand during intense physical exertion. Conversely, significant changes in the RBC, Hgb, and Hct levels were not observed, possibly due to the short duration of the match and the athletes’ specific conditioning levels. Future studies should include longer recovery periods to better evaluate the comprehensive impact of kickboxing K1 matches on athletes’ hematological parameters.

## Figures and Tables

**Table 1 jfmk-09-00130-t001:** Anthropometric and demographic information of participants.

	Max	Min	$\bar{x}$ ± SD
Age (years)	28	19	23.71 ± 3.82
Body Height (cm)	184	176	179.6 ± 3.11
Body weight (kg)	84	71	79.2 ± 5.54

$\bar{x}$—arithmetic mean; SD—standard deviation; Min—minimum value; Max—maximum value.

**Table 2 jfmk-09-00130-t002:** Changes in hematological parameters of participants following K1 kickboxing match.

Variable	Test	n	$\bar{x}$	SD	t	df	*p*
WBC (10^9^/L)	Pre	10	6.49	1.91	−10.838	9	0.000 *
	Post	10	11.17	2.42			
Plt (10^9^/L)	Pre	10	261.00	28.19	−13.450	9	0.000 *
	Post	10	334.20	39.32			
Neut (%)	Pre	10	61.83	7.36	7.304	9	0.000 *
	Post	10	50.92	9.13			
Lymph (%)	Pre	10	26.03	6.51	−8.063	9	0.000 *
	Post	10	37.74	9.22			
Mono (%)	Pre	10	0.64	0.24	−6.748	9	0.000 *
	Post	10	1.04	0.35			
RBC (10^12^/L)	Pre	10	5.25	0.30	0.722	9	0.488
	Post	10	5.20	0.23			
Hgb (g/dL)	Pre	10	15.66	0.70	0.670	9	0.520
	Post	10	15.55	0.46			
Hct (%)	Pre	10	46.65	2.02	−1.251	9	0.243
	Post	10	47.45	1.63			
CK (U/L)	Pre	10	288.10	187.84	−4.036	9	0.000 *
	Post	10	330.30	206.58			
La (mmol/L)	Pre	10	1.94	0.35	−4.04	9	0.003 *
	Post	10	10.23	1.05			
Glucose (mg/dL)	Pre	10	91.900	8.646	−10.205	9	0.000 *
	Post	10	168.500	27.504			

n—number of respondents; $\bar{x}$—arithmetic mean; SD—standard deviation; t—value for trial pairs; df—degrees of freedom; *p*—level of significance of variation; WBC—white blood cells count; Plt—platelets; Neut—neutrophil content; Lymph—lymphocytes; Mono—monocytes; RBC—red blood cells count; Hgb—hemoglobin; Hct—hematocrit; CK—creatine kinase activity; La—lactate concentration. * *p* < 0.05

## Data Availability

The data presented in this study are available upon request from the corresponding author.

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
