# Peer review of "Acute Effects of Kickboxing K1 Matches on Hematological Parameters of Kickboxers"

_jfmk, 2024, doi:10.3390/jfmk9030130_

Round 1
Reviewer 1 Report
Comments and Suggestions for Authors
The authors aimed to evaluate the acute changes in hematological parameters, glucose and lactic acid (or lactate concentration?) in kickboxers following an official kick boxing K1 match.
The subject of this work is interesting, but the manuscript in its current form is not suitable for publication.
First of all the number of athletes is too small and can influence the statistical analysis. When determining the minimum sample size, it is recommended to use statistical tools (for example G*Power).
Moreover, the description of blood sampling and biochemical tests is incorrect. Are you sure that the analyzes were performed in serum? Automatic hematology analyzers (like Sysmex) determine morphotic elements in whole blood, not serum. Sysmex K-1000 did not assess glucose, lactic acid, or creatine kinase. Serum lactate (La), not lactic acid (LA), can be tested.
Abbreviations used in the manuscript should be changed (Lymphocytes - Lym, hematocrit – Hct etc.).
The abstract should include statistical methods and p-values to support the results.
Line 98: Samples taken 2 minutes before the match do not indicate a resting value. Before the match, the athletes probably performed a warm-up, which influenced the tested parameters. So these were rather pre-match values.
Line 255: In the cited bibliography, the concentration of lactate, not lactic acid, was tested.
Line 271: In the discussion chapter, the authors should add a paragraph on creatine kinase to be able to draw conclusions on this basis.
Comments on the Quality of English LanguageAuthors should perform a general review of the text in terms of vocabulary and verbal adequacy. The text is relatively clear and sufficient for the reader to understand it.
Author Response
Dear Reviewer,
Your suggestions have been invaluable to us. Following your recommendations, we have made some changes to our research paper. All modifications to the text have been highlighted in yellow. You can find detailed information in the attached report. We sincerely appreciate your contributions.
Best regards,
“The authors aimed to evaluate the acute changes in hematological parameters, glucose and lactic acid (or lactate concentration?) in kickboxers following an official kick boxing K1 match.”
The relevant expression has been replaced with the expression lactate concentration.
“First of all the number of athletes is too small and can influence the statistical analysis. When determining the minimum sample size, it is recommended to use statistical tools (for example G*Power).”
This section has been updated as follows:
“The minimum sample size for this study was calculated using G*Power software 3.1.9.7. (Dusseldorf University, Dusseldorf, Germany). Accordingly, T-tests were utilized to calculate power according to the study design; Means: difference between two dependent means; α error probability = 0.05; effect size = 0.90 and power (1-β error probability) = 0.80 were determined. Based on this analysis the minimum sample size required for statistical significance as determined by the software with a real power of 0.83% was understood to be at least 10 participants. Therefore, the participant group of this study includes 10 kickboxers. The participant group of this study was determined using the homogeneous sampling technique within the scope of the purposeful sampling method.”
“Moreover, the description of blood sampling and biochemical tests is incorrect. Are you sure that the analyzes were performed in serum? Automatic hematology analyzers (like Sysmex) determine morphotic elements in whole blood, not serum. Sysmex K-1000 did not assess glucose, lactic acid, or creatine kinase. Serum lactate (La), not lactic acid (LA), can be tested.”
This section has been updated as follows:
“Venous blood samples were collected from participants by an experienced spe-cialist using the venipuncture method 2 minutes before the match (resting level) and 2 minutes after the match (acute effect of the match). A volume of 5 ml of blood was drawn during each blood collection process. The collected blood samples were trans-ferred to EDTA tubes for hematological analysis and biochemistry tubes for serum separation, maintaining cold chain transportation to ensure sample integrity during transit to the laboratory.
For hematological analysis, the blood samples in EDTA tubes were analyzed using a Sysmex automatic hematology analyzer (Model K-1000, Sysmex Corporation, Japan), which determines morphotic elements in whole blood, including White Blood Cells (WBC), Platelets (PLT), Neutrophils (NEUT), Lymphocytes (LYMPH), Monocytes (MONO), Red Blood Cells (RBC), Hemoglobin (HGB), and Hematocrit (HCT).
The blood samples in biochemistry tubes were centrifuged at 4000 rpm for 10 minutes at +4°C to separate the serum. The serum samples obtained were then stored at -40°C until biochemical analyses were performed. When the time for analysis ar-rived, the samples were thawed at room temperature. Serum levels of Creatine Kinase (CK), Lactate (La), and Glucose were measured using appropriate biochemical analyz-ers.”
“Abbreviations used in the manuscript should be changed (Lymphocytes - Lym, hematocrit – Hct etc.).”
Could you please elaborate your suggestions about the abbreviation form of the words.
“The abstract should include statistical methods and p-values to support the results.”
The statement (p<0.05) has been added to the relevant section.
Additionally, the following statement has been added to the relevant place.
“Paired sample t-tests were used to compare the pre-test and post-test hematological parameters of the participants.”
“Line 98: Samples taken 2 minutes before the match do not indicate a resting value. Before the match, the athletes probably performed a warm-up, which influenced the tested parameters. So these were rather pre-match values.”
The phrase "resting level" has been removed from the text.
“Line 255: In the cited bibliography, the concentration of lactate, not lactic acid, was tested.”
The relevant expression has been replaced with the expression lactate concentration.
“Line 271: In the discussion chapter, the authors should add a paragraph on creatine kinase to be able to draw conclusions on this basis.”
The following paragraph has been added to the discussion section.
“As the CK molecule is one of the most common serum markers of muscle injury [60,61], the statistical changes observed in CK levels as a result of our findings indicate that a K1 kickboxing competition induces muscle damage in athletes. The literature contains similar studies supporting this finding. In this context, Poderoso de Souza et al. [62] reported an increase in CK levels in athletes following an MMA fight. Addi-tionally, these findings are consistent with the observations by Brandão et al. [63], who noted an increase in CK levels in athletes after a Jiu-Jitsu competition. The rise in CK levels reflects muscle damage in active muscles, indicating that the integrity and composition of the plasma membrane are affected [64].”
Reviewer 2 Report
Comments and Suggestions for Authors
I first would like to congratulate the author for this interesting study about assessment of hematological parameters of kickboxers. It’s a considerable effort from the researchers and the materials involved.
It’s well written and follow the IMRAD design and describes adequately the content of the research protocol. The background of the study is clear and points out the need to further research in this field. The methods and fundings are clear, and well justified as well as the methodology regarding risk of bias analysis and statistic techniques. Moreover, the results and discussion are well written, and present and discuss the main results of the research in adequate manner, clearly presenting the main outcomes of the research, and discussing them accordingly.
After reading I’ d like to suggest some adjustments, some improvements, and corrections.
There are several typos of hyphenation. Please, revised the entire text.
Although the introduction is sufficiently clear, I would suggest explaining in a more comprehensive and detailed way the background of the study (i.e., why authors decided to conduct this study, possible methodological implications for sports science, effects on short-medium-long terms health status, etc.).
I would suggest explaining in more detail the procedure by which the sample was recruited (simple randomization, convenience sample, etc.).
Moreover, according to the first comment, I recommend explaining why authors used hematological parameters and not other indicators (I think about cortisol levels for stress, etc.), as well as the reasons why assessment of hematological parameters was conducted two minutes after the match.
Also the results should be described in a more detailed way.
Considering the non-homogeneous sample and the sample’s source, I strongly recommend adding a separate paragraph about study limitations (i.e., sample size, possible covariates not considered, the choice of hematological parameters, acute effects of exercise, etc.) and methodological implications in sports sciences (i.e., recovery, intensity of exercise, stress, etc.).
I would suggest these major changes, and I would like to congratulate the author for the important effort in doing this research and acknowledge the contribution for the understanding of the topic.
Comments on the Quality of English LanguageThere are several typos of hyphenation. Please, revised the entire text.
Author Response
Dear Reviewer,
Your suggestions have been invaluable to us. Following your recommendations, we have made some changes to our research paper. All modifications to the text have been highlighted in yellow. You can find detailed information in the attached report. We sincerely appreciate your contributions.
Best regards,
“There are several typos of hyphenation. Please, revised the entire text.”
The entire text was checked and relevant changes were made.
“Although the introduction is sufficiently clear, I would suggest explaining in a more comprehensive and detailed way the background of the study (i.e., why authors decided to conduct this study, possible methodological implications for sports science, effects on short-medium-long terms health status, etc.).”
Why this research is important is emphasized by adding the following statements to the introductory text. “Understanding the effects of an official Kickboxing K1 match on certain blood parameters of athletes can facilitate the development of more effective approaches in training programs and pre- and post-competition care strategies. Additionally, it can provide significant clinical guidance for kickboxing coaches, sports physicians, and other relevant health professionals, thereby enhancing athlete health and performance. This study also offers a comprehensive framework for understanding the ex-tent of physiological stress experienced by athletes, which could benefit other athletes subjected to similar intense physical activity.”
“I would suggest explaining in more detail the procedure by which the sample was recruited (simple randomization, convenience sample, etc.).”
This section has been updated as follows:
“The minimum sample size for this study was calculated using G*Power software 3.1.9.7. (Dusseldorf University, Dusseldorf, Germany). Accordingly, T-tests were utilized to calculate power according to the study design; Means: difference between two dependent means; α error probability = 0.05; effect size = 0.90 and power (1-β error probability) = 0.80 were determined. Based on this analysis the minimum sample size required for statistical significance as determined by the software with a real power of 0.83% was understood to be at least 10 participants. Therefore, the participant group of this study includes 10 kickboxers. The participant group of this study was determined using the homogeneous sampling technique within the scope of the purposeful sampling method.”
“Moreover, according to the first comment, I recommend explaining why authors used hematological parameters and not other indicators (I think about cortisol levels for stress, etc.), as well as the reasons why assessment of hematological parameters was conducted two minutes after the match.”
Dear Reviewer, We fully acknowledge and agree with your suggestion. However, we would like to remind you that conducting scientific research sometimes requires a certain amount of funding. The funding for our research was only sufficient to test the current parameters, and therefore, we were unable to examine parameters such as cortisol etc.
Additionally, the fact that blood samples were taken 2 minutes after the competition is due to our concern to determine the acute effect.
“Also the results should be described in a more detailed way.”
Dear Reviewer, When reporting research findings, we preferred to emphasize only statistically significant differences, which may create an impression that the research findings were not adequately explained. We discussed our interpretations of the findings in the discussion section, providing a detailed analysis of the research findings.
“Considering the non-homogeneous sample and the sample’s source, I strongly recommend adding a separate paragraph about study limitations (i.e., sample size, possible covariates not considered, the choice of hematological parameters, acute effects of exercise, etc.) and methodological implications in sports sciences (i.e., recovery, intensity of exercise, stress, etc.).”
Details regarding the number and determination of the participants of the study were re-done in the method section of the research. Other limitations are clearly stated in the discussion section of the study.
Reviewer 3 Report
Comments and Suggestions for Authors
Introduction
Line 30: Regarding the demands of kickboxing, even though you mentioned strength, I recommend emphasizing muscle power.
Line 38: Standardize Kickboxing writing. There are times when it is cited separately.
The introduction has several words associated with "-", such as: in-tensity, understand-ing, fol-lowuing. Adjust all these words.
The introduction needs improvement. Cite the reasons for the study. They only mention that there are limited studies on the impact of kickboxing on hematological parameters. Build a more solid justification on the subject. Also, what were the study's hypotheses? They did not.
Methods
First they should mention the participants, then the experimental design.
They mentioned that they evaluated lactic acid. But it seems to me that this evaluation was about the behavior of lactate. They are different components, please explain that.
Discussion
Line 141: They cited findings in soccer players. For me, it doesn't make sense to use these findings to justify those of the present study. In addition, half-marathons, basketball and judo were cited. The interesting thing is to look for studies on boxing, taekwondo, karate... sports with the same motor gesture characteristics. If you can't, use the argument about these difficulties. It will be better than comparing studies with sports that are totally different in terms of motor skills.
Line 205 / 206: About RBC, HCT and HGB. The information on these components has been extrapolated. Since there was a slight difference between them: for CBR: 0.05 ml.ul; HCT: 0.8%; HGB: 0.11 g/dl. These differences are insignificant to assume the above. I suggest rewriting that part.
The study made some very important analyses, but in general, what is its outcome for practical applicability? What can these findings do for coaches and athletes? What is the relevance of these findings, even if only acutely, in order to consider the study of great importance to the sport of kickboxing? The authors need to answer these questions in the discussion, clarifying these issues
With regard to limitations, the number of participants (which could be greater) and studies to discuss the findings (if they don't find the ones recommended).
Conclusion
The first sentence (lines 269 and 270) should go into the discussion. This is the kind of information that doesn't fit with the conclusion.
Regarding CK, the time of collection (2 minutes after exertion) is not the ideal time to assess this component. Therefore, I recommend changing the wording so as not to extrapolate what was found.
Comments on the Quality of English LanguageI recommend proofreading your writing. Many words are misspelled, for example separated by "-".
Author Response
Dear Reviewer,
Your suggestions have been invaluable to us. Following your recommendations, we have made some changes to our research paper. All modifications to the text have been highlighted in yellow. You can find detailed information in the attached report. We sincerely appreciate your contributions.
Best regards,
Introduction
“Line 30: Regarding the demands of kickboxing, even though you mentioned strength, I recommend emphasizing muscle power.”
The expression strength has been replaced by the expression muscle power.
“Line 38: Standardize Kickboxing writing. There are times when it is cited separately.”
The relevant word is now standardized as Kickboxing throughout the text.
“The introduction has several words associated with "-", such as: in-tensity, understand-ing, fol-lowuing. Adjust all these words.”
Relevant corrections have been made.
“The introduction needs improvement. Cite the reasons for the study. They only mention that there are limited studies on the impact of kickboxing on hematological parameters. Build a more solid justification on the subject. Also, what were the study's hypotheses? They did not.”
Why this research is important is emphasized by adding the following statements to the introductory text. “Understanding the effects of an official Kickboxing K1 match on certain blood parameters of athletes can facilitate the development of more effective approaches in training programs and pre- and post-competition care strategies. Additionally, it can provide significant clinical guidance for kickboxing coaches, sports physicians, and other relevant health professionals, thereby enhancing athlete health and performance. This study also offers a comprehensive framework for understanding the ex-tent of physiological stress experienced by athletes, which could benefit other athletes subjected to similar intense physical activity.”
The hypothesis of the research has been added to the text as stated below. “The primary hypothesis of this research was that an official Kickboxing K1 match will induce acute changes in the blood parameters of the athletes.”
Methods
“First they should mention the participants, then the experimental design.”
The participants section was placed before the design of the study.
“They mentioned that they evaluated lactic acid. But it seems to me that this evaluation was about the behavior of lactate. They are different components, please explain that.”
The subject was clarified by using the expression " lactate concentration " in the text and it was ensured that no misunderstanding was caused.
Discussion
“Line 141: They cited findings in soccer players. For me, it doesn't make sense to use these findings to justify those of the present study. In addition, half-marathons, basketball and judo were cited. The interesting thing is to look for studies on boxing, taekwondo, karate... sports with the same motor gesture characteristics. If you can't, use the argument about these difficulties. It will be better than comparing studies with sports that are totally different in terms of motor skills.”
In the discussion section of this study, we have removed references to studies on football, basketball, and rugby as per your suggestion. However, we believe it would be meaningful to include findings from previous research on judo, as it is also a combat sport. Also we emphasized that similar studies in the field are quite limited.
“Line 205 / 206: About RBC, HCT and HGB. The information on these components has been extrapolated. Since there was a slight difference between them: for CBR: 0.05 ml.ul; HCT: 0.8%; HGB: 0.11 g/dl. These differences are insignificant to assume the above. I suggest rewriting that part.”
The relevant sentence has been modified as you suggested.
“The study made some very important analyses, but in general, what is its outcome for practical applicability? What can these findings do for coaches and athletes? What is the relevance of these findings, even if only acutely, in order to consider the study of great importance to the sport of kickboxing? The authors need to answer these questions in the discussion, clarifying these issues”
The following paragraph regarding this subject has been added to the text. “Coaches and athletes can use these findings to better understand the immediate physiological responses of kickboxers post-competition. This knowledge can aid in op-timizing recovery strategies and adjusting training programs to mitigate potential negative impacts on health and performance. Knowing that certain hematological pa-rameters like WBC, PLT, and glucose levels increase significantly post-match, coaches can tailor training sessions and recovery protocols accordingly. For instance, empha-sizing recovery strategies that address these acute changes, such as nutrition and hy-dration strategies, can potentially enhance performance and reduce injury risks.”
“With regard to limitations, the number of participants (which could be greater) and studies to discuss the findings (if they don't find the ones recommended).”
The limitations of the research section has been re-detailed taking your suggestions into consideration.
Conclusion
“The first sentence (lines 269 and 270) should go into the discussion. This is the kind of information that doesn't fit with the conclusion.
Regarding CK, the time of collection (2 minutes after exertion) is not the ideal time to assess this component. Therefore, I recommend changing the wording so as not to extrapolate what was found.”
The conclusion section has been updated as follows.”This study observed significant increases in WBC, PLT, NEUT, and LYMPH levels following K1 matches, indicating substantial physiological responses among kickboxers. Additionally, elevated CK levels were recorded, suggesting potential muscle damage and early stages of repair. However, it is important to note that assessing CK levels shortly after exertion (2 minutes) may not accurately reflect peak levels. Increases in LA levels reflect heightened anaerobic metabolism, while elevated glucose levels likely indicate increased energy demand during intense physical exertion. Conversely, significant changes in RBC, HGB, and HCT levels were not observed, possibly due to the short duration of matches and athletes' specific conditioning levels. Future studies should include longer recovery periods to better evaluate the comprehensive impact of kickboxing K1 matches on athletes' hematological parameters.”
Round 2
Reviewer 1 Report
Comments and Suggestions for Authors
The authors should read the manuscript carefully to eliminate editorial and linguistic errors. The discussion is currently difficult to understand. Perhaps adding subtitles will help organize the discussion. As can be read in the instructions for authors: “Authors should discuss the results and how they can be interpreted from the perspective of previous studies and of the working hypotheses. The findings and their implications should be discussed in the broadest context possible.”
The authors use the terms lactic acid (LA) and lactate (La) as synonyms. Please standardize throughout the manuscript.
Abbreviations used in the manuscript should be changed according to the following pattern:
White Blood Cells – WBC, Red Blood Cells – RBC, Platelets – Plt, Lymphocytes – Lymph (authors use LYMPH and LYM - please standardize), Hematocrit – Hct etc.).”
Line 49: should be “red blood cells” not red blood cell
Line 137: Please describe “appropriate biochemical analyzers”
There are errors in units in Table 2. Please also correct abbreviations.
Line 180: “The increase in WBC concentration following exercise can be attributed, in part, to the reduction in plasma volume.” – Please describe a phenomenon of reduction in plasma volume during short-term exercise. How to explain the potential change in plasma volume without changing hematocrit?
Line 242: “Wardyn et al., [47] reported a significant increase in the concentration of HTC and HGB. [8] reported that high intensity acute judo exercise led to a decrease in HCT levels in adolescent elite judokas.” - unclear sentence
Line 267: “The immediate decrease in RBCs post-exercise can be attributed to short-term hemoconcentration induced by plasma loss during the match” - How to explain the potential change in plasma volume without changing hematocrit?
Line 344: “Data Availability Statement: The data presented in this study are available upon request from the corresponding author.” - The results should be made available to verify any calculation errors (for example in the research data repository).
Comments on the Quality of English LanguageMinor editing of English language required.
Author Response
The authors should read the manuscript carefully to eliminate editorial and linguistic errors. The discussion is currently difficult to understand. Perhaps adding subtitles will help organize the discussion. As can be read in the instructions for authors: “Authors should discuss the results and how they can be interpreted from the perspective of previous studies and of the working hypotheses. The findings and their implications should be discussed in the broadest context possible.”
Dear reviewer, thank you very much for your comments and suggestions on our research. we would like you to know that we have carefully made all possible corrections. the relevant corrections to the text are set out in more detail below. thank you again for your contribution to our research.
The authors use the terms lactic acid (LA) and lactate (La) as synonyms. Please standardize throughout the manuscript.
In the text, the relevant expression is standardized as lactate (La).
Abbreviations used in the manuscript should be changed according to the following pattern:
White Blood Cells – WBC, Red Blood Cells – RBC, Platelets – Plt, Lymphocytes – Lymph (authors use LYMPH and LYM - please standardize), Hematocrit – Hct etc.).”
Relevant corrections have been made in the text.
Line 49: should be “red blood cells” not red blood cell
The phrase red blood cell has been changed to red blood cells.
Line 137: Please describe “appropriate biochemical analyzers”
Analyzer information added (Beckman Coulter AU480, USA).
There are errors in units in Table 2. Please also correct abbreviations.
The abbreviations have been corrected, but we could not detect any errors in the units. Please specify any errors in detail, if any.
Line 180: “The increase in WBC concentration following exercise can be attributed, in part, to the reduction in plasma volume.” – Please describe a phenomenon of reduction in plasma volume during short-term exercise. How to explain the potential change in plasma volume without changing hematocrit?
This paragraph has been supported by adding the following statement. “It is thought that the decrease in plasma volume during short-term exercise may have occurred due to the shift of fluid into the interstitial spaces, increased sweating and elevated blood pressure, which may concentrate blood components such as white blood cells in bulk.”
Line 242: “Wardyn et al., [47] reported a significant increase in the concentration of HTC and HGB. [8] reported that high intensity acute judo exercise led to a decrease in HCT levels in adolescent elite judokas.” - unclear sentence
The sentence was corrected by adding author information.
Line 267: “The immediate decrease in RBCs post-exercise can be attributed to short-term hemoconcentration induced by plasma loss during the match” - How to explain the potential change in plasma volume without changing hematocrit?
This paragraph has been supported by adding the following statement. “The potential change in plasma volume without altering hematocrit can be explained by the immediate release of stored red blood cells from the spleen, which maintains the ratio of red blood cells to plasma.”
Line 344: “Data Availability Statement: The data presented in this study are available upon request from the corresponding author.” - The results should be made available to verify any calculation errors (for example in the research data repository).
Dear reviewer, we understand your suggestion on this matter, but the fact that the data can be provided by the corresponding author upon request also makes the data available.
Reviewer 2 Report
Comments and Suggestions for Authors
Appropriate revisions have been made.
Author Response
Thank you
Reviewer 3 Report
Comments and Suggestions for Authors
The authors have made the necessary adjustments and need to leave the text in the format of the final version for publication.
Author Response
Thank you , I am sending the version with the accepted changes in PDF
Round 3
Reviewer 1 Report
Comments and Suggestions for Authors
There are errors in units in Table 2. I suggest changing the WBC, Plt and RBC units according to the formula below:
WBC (109/L)
RBC (1012/L)
Plt (109/L)
Author Response
Dear Reviewer,
We sincerely thank you for your meticulous evaluation and detailed feedback on our study. Your suggestions and criticisms have significantly contributed to strengthening and enhancing the value of our paper. As per your latest revision recommendations, we have made the necessary adjustments and improvements. We are grateful for your efforts in improving the scientific merit of our work. Thank you once again for dedicating your valuable time to reviewing our manuscript.
Sincerely.